# Rapidly Progressive Metastatic Angiosarcoma of the Heart: A Case Report

**DOI:** 10.3390/diagnostics13162666

**Published:** 2023-08-13

**Authors:** Soo Yeon An, Man-Shik Shim

**Affiliations:** 1Department of Medical Sciences, College of Medicine, Chungnam National University, 266 Moonhwa-lo, Daejeon 35015, Republic of Korea; 2Division of Cardiology, Department of Internal Medicine, Chungnam National University Hospital, 282 Moonhwa-lo, Daejeon 35015, Republic of Korea; 3Department of Cardiothoracic Surgery, Chungnam National University Hospital, 282 Moonhwa-lo, Daejeon 35015, Republic of Korea; mdshmsh@naver.com

**Keywords:** cardiac mass, primary angiosarcoma, multimodality imaging, cardiac magnetic resonance imaging

## Abstract

Cardiac angiosarcoma is a rare, malignant neoplasm affecting the heart. We present the clinical history of a 40-year-old patient diagnosed with metastatic angiosarcoma of the heart. The patient complained of shortness of breath, and a cardiac computed tomography scan revealed a mass in the right atrium and pericardial effusion. Transthoracic and transesophageal echocardiography provided detailed anatomical and functional information, and cardiac magnetic resonance imaging and fluorodeoxyglucose-positron emission tomography/computed tomography were used to assess distant metastasis and characterize the mass in detail. Early differential diagnosis and comprehensive evaluation of a cardiac mass are vital for determining appropriate treatment strategies to improve patient outcomes. The pathological results from the endocardial biopsy confirmed the diagnosis of primary angiosarcoma. The patient underwent surgical resection of the right atrial mass but died within one month because of the locally advanced nature of the angiosarcoma and its rapid progression. The patient’s medical journey sheds light on the challenges associated with diagnosing and treating this rare condition, particularly the rapid progression of its cardiac manifestations.

Primary heart tumors are relatively rare, with autopsies of adult patients revealing a low incidence rate. The prevalence of primary heart tumors ranges from 0.001% to 0.03% [1]. Although less frequent, malignant heart tumors pose significant challenges in diagnosis and treatment because of their aggressive nature. Primary cardiac angiosarcoma, a malignancy originating from the endothelial cells, is the most common primary differentiated cardiac neoplasm [2]. It predominantly affects individuals aged 40–50 years and can develop in any heart chamber [3]. Here, we present a case of a 40-year-old man diagnosed with primary cardiac angiosarcoma.

**Figure 1 diagnostics-13-02666-f001:**
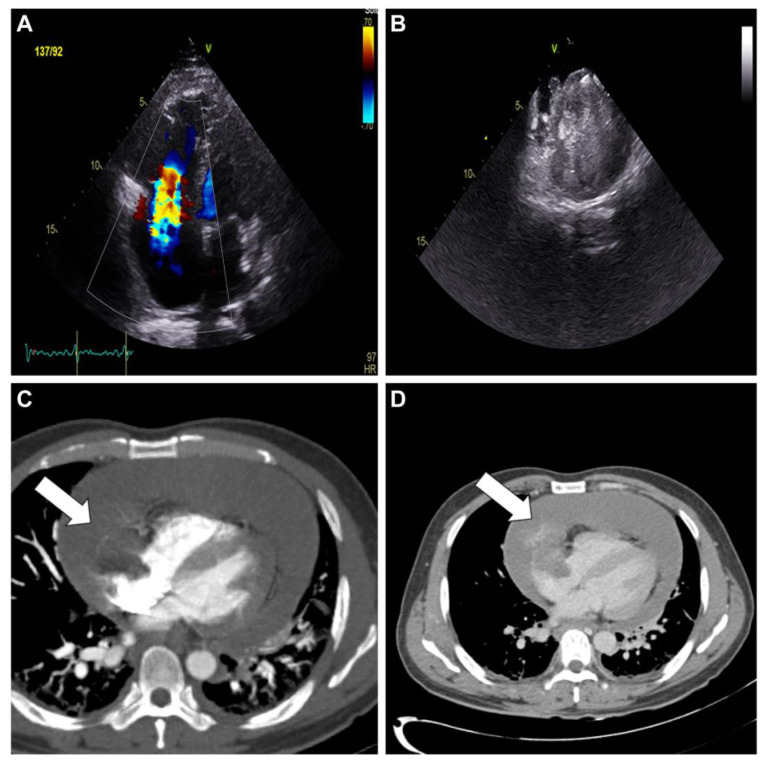
Initial evaluation using transthoracic echocardiography demonstrated a preserved ejection fraction of the left ventricle, a dilated right ventricle with a moderate degree of tricuspid regurgitation (**A**), and mild tricuspid stenosis on color Doppler, in addition to a significant amount of pericardial effusion, measuring 23 mm at the apex. The patient underwent pericardiocentesis to alleviate the symptoms. Approximately 200 mL of sanguineous fluid was extracted from the catheter, and microbiological and cytological examinations were negative for infection or malignancy. Subsequent transesophageal echocardiography revealed a hyperechoic mass in the right atrium. The mass protruded through the tricuspid valve and deeply penetrated the posterior wall of the right atrium (**B**). Further investigation using cardiac computed tomography with contrast enhancement confirmed the presence of a large mass (white arrow)in the right atrium sized 42 mm × 35 mm in the axial plane (**C**) and inhomogeneous central enhancement in the delayed phase, indicating a vascular mass (white arrow) with hemopericardium (**D**).

**Figure 2 diagnostics-13-02666-f002:**
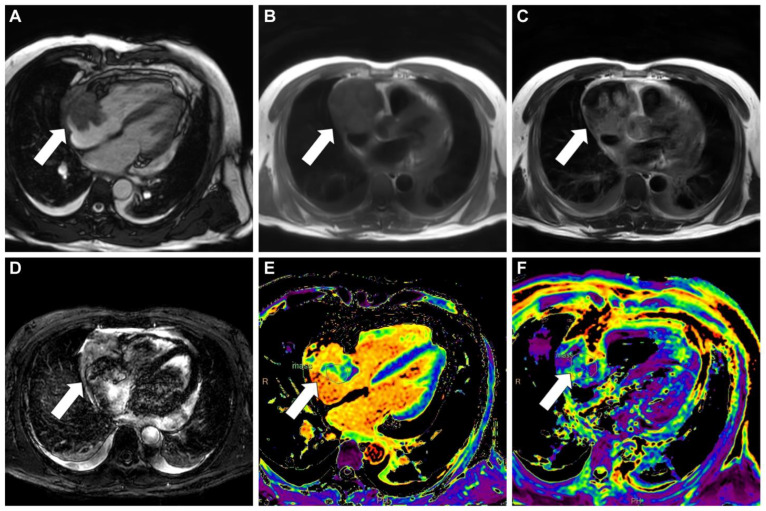
A cardiac magnetic resonance scan revealed an immobile mass (white arrow) in the right atrium extending to the pericardial space (**A**). The mass appeared predominantly isointense on T1-weighted images (**B**) and heterogeneous on T2-weighted images (**C**). The hemorrhagic areas within the mass exhibited increased signal intensity on T2-weighted short tau inversion recovery images (**D**). T1/T2 mapping revealed a long T1/long T2 profile (**E**,**F**). Histopathological examination of the endocardial biopsy of the right atrial mass was consistent with primary angiosarcoma showing high mitotic activity (39/10 high power field) and necrosis in hematoxylin and eosinophil stain. Immunohistochemical stain for specific tumor markers was positive for CD31, ERG, and CD34.

**Figure 3 diagnostics-13-02666-f003:**
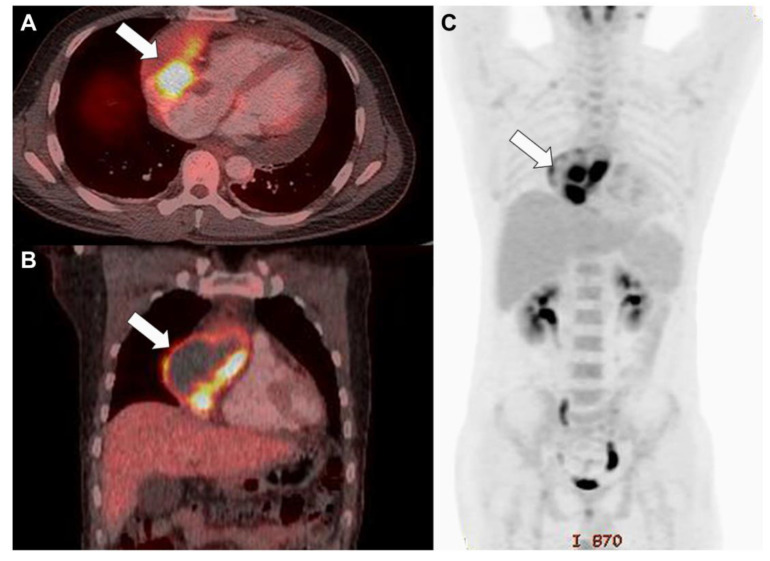
Fluorodeoxyglucose-positron emission tomography/computed tomography indicated increased uptake in the mass (white arrow) and a hypointense region filled with fluid in the axial (**A**) and coronal sections (**B**) and direct invasion into the great vessels without evidence of distant metastasis (**C**).

**Figure 4 diagnostics-13-02666-f004:**
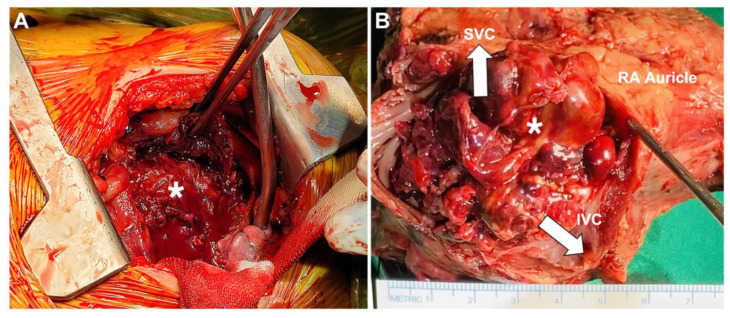
The patient underwent surgical resection of the right atrial mass (asterisk) and the malignant tissue in the pericardium and epicardium. Gross examination during surgery revealed an oval-shaped mass, 59 mm × 75 mm × 75 mm in size, occupying the right atrium (**A**) and invading the surrounding great vessels (white arrows) and pericardium (**B**).

Although the initial surgery provided temporary relief, the infiltrative nature of cardiac angiosarcoma complicated complete tumor removal, and the patient’s condition continued to deteriorate. This necessitated a critical decision for a second surgical procedure. The medical team opted for a window operative surgery, a technique aimed at creating a controlled opening in the pericardial sac to release pressure on the heart and allow for tumor excision. The tumor’s rapid growth outpaced the surgical interventions, leading to further exacerbation of cardiac tamponade. The patient died three days after the second surgery. From this case report, we could learn that early differential diagnosis using multimodal imaging is crucial for the optimal treatment of rapidly progressive cardiac malignancies. Cardiac tumors can present with a wide range of symptoms, making diagnosis and characterization challenging during the initial diagnostic process. The tumor can obstruct blood flow within the heart or interfere with valve function. However, local invasion can lead to arrhythmia or fluid accumulation in the pericardial sac with potential tamponade [4]. Chest radiography and transthoracic echocardiography (TTE) have limited diagnostic efficacy in identifying cardiac masses. TTE is often the initial imaging modality used; however, its limitations, such as dependence on a favorable window and limited tissue characterization, necessitate the use of additional techniques [5]. Transesophageal echocardiography and contrast-enhanced computed tomography serve as adjunctive measures to aid the visualization of cardiac masses [6]. Although CT provides valuable insights into coronary artery anatomy and calcification, its reliance on ionizing radiation and limitations in myocardial tissue characterization restrict its applicability. Cardiac magnetic resonance imaging (CMR) and fluorodeoxyglucose-positron emission tomography/computed tomography (FDG-PET/CT) are advanced imaging modalities used for the comprehensive characterization of cardiac masses, assessment of local invasion, and evaluation of distant metastasis [7]. FDG-PET/CT, with its metabolic insights, complements structural imaging but faces challenges related to radiotracer availability and spatial resolution. Accordingly, CMR’s capability of comprehensive assessment, lack of ionizing radiation, and superior soft tissue resolution render it invaluable for a wide range of cardiac pathologies, ranging from benign lesions to malignant tumors. CMR has emerged as the reference standard for assessing suspected cardiac tumors, offering a comprehensive evaluation of the size, shape, attachment, relationship to surrounding structures, hemodynamic impact, and tissue characterization [8]. CMR’s versatility in providing T1-weighted and T2-weighted images using various pulse sequences, including balanced steady-state free precession cine and short tau inversion recovery, enables accurate diagnosis and treatment planning [9]. Additionally, native T1 and T2 mapping have emerged as an adjunctive technique for further tissue-characterizing cardiac tumors [10]. Integrating multiple imaging modalities plays a crucial role in overcoming the limitations of individual techniques, providing a comprehensive evaluation, and guiding optimal management strategies for patients with cardiac masses. CMR stands out as an imaging modality with remarkable capabilities for resolving diagnostic dilemmas associated with cardiac masses. Employing CMR at an earlier stage in the diagnostic process can enhance clinical decision making and patient outcomes.

## Data Availability

Not applicable as no new data were generated.

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
