# Peer review of "Rapidly Progressive Metastatic Angiosarcoma of the Heart: A Case Report"

_diagnostics, 2023, doi:10.3390/diagnostics13162666_

Round 1

Reviewer 1 Report

Please give an introduction. The picture should not be the first what a person sees.Introduction begins with 35! Than 38 begin with pictute and what you want to describe to the pictures.one should see what ist normal text and what is Fig. text.same with fig.4- it ends at 65 than begin the normal text with  THE PATIENT....

To my opinion CT is inferior to MR state this clearly. PET/CT is of limited value- yes you see there is no metastasis. But you can do whole body MR - without radiation and the same result - no metastases. What you need is an TWIST sequence oder flash 3 D contrast enhanced MRA to clerly show the vessels and the invasion. Here you can discuss a CTA of coronaries - sometimes plays a role.

As discussionof the case: how can we earlir detect such tumors? it was to late in your patient!! So say, as like mammography every second year do MR of whole body in all people to see early the masses - also renal or pancreatic .. or such rare as well!

This should be the AIM  - to help the next generation!You can give a forward tip: prospective study of 10 000 people of a region with whole body MR every 2 yers and a population without - what do we see - early masses- where! Do not hesitate to give hint for imging: MR agiography and diffusion of whole body should be part of the screening of every person. WE DO NOT NEED CT OR THE OTHER METHODS AT FIRST!One sequece with fat suppression, mapping as you showin fig 2 E and F is nice to have but not essential!

best regards

Author Response

Response to Reviewer 1 Comments

Point 1: Please give an introduction. The picture should not be the first what a person sees.Introduction begins with 35! Than 38 begin with pictute and what you want to describe to the pictures.one should see what ist normal text and what is Fig. text.same with fig.4- it ends at 65 than begin the normal text with  THE PATIENT...

Response 1: I will add an introduction and separate the normal texts from the figure legends of Figure 4 according to your suggstions.

Point 2: To my opinion CT is inferior to MR state this clearly. PET/CT is of limited value- yes you see there is no metastasis. But you can do whole body MR - without radiation and the same result - no metastases. What you need is an TWIST sequence oder flash 3 D contrast enhanced MRA to clerly show the vessels and the invasion. Here you can discuss a CTA of coronaries - sometimes plays a role.

Response 2: I have added a descripition describing the limitations of CT and PET/CT compared to MRI according to your opinions.

(Line 86-88)

Although CT provides valuable insights into coronary artery anatomy and calcification, its reliance on ionizing radiation and limitations in myocardial tissue characterization restrict its applicability.

(Line 91-95)

FDG-PET/CT, with its metabolic insights, complements structural imaging but faces challenges related to radiotracer availability and spatial resolution. Accordingly, CMR's comprehensive assessment capability, lack of ionizing radiation, and superior soft tissue resolution render it invaluable for a wide range of cardiac pathologies, ranging from benign lesions to malignant tumors.

Point 3:  As discussion of the case: how can we earlir detect such tumors? it was to late in your patient!! So say, as like mammography every second year do MR of whole body in all people to see early the masses - also renal or pancreatic .. or such rare as well! This should be the AIM  - to help the next generation!You can give a forward tip: prospective study of 10 000 people of a region with whole body MR every 2 yers and a population without - what do we see - early masses- where! Do not hesitate to give hint for imging: MR agiography and diffusion of whole body should be part of the screening of every person. WE DO NOT NEED CT OR THE OTHER METHODS AT FIRST!One sequece with fat suppression, mapping as you showin fig 2 E and F is nice to have but not essential!

Response 3: I have added a suggesion for detecting caridac masses in early stages.

(Line 104-106)

CMR stands out as an imaging modality with remarkable capabilities for resolving diagnostic dilemmas associated with cardiac masses. Employing CMR at an earlier stage in the diagnostic process can enhance clinical decision-making and patient outcomes.

Reviewer 2 Report

The paper presents a rare however very important issue of angiosarcoma of the heart. There is scarce information on the diagnostics and treatment of such rare cases in the literature, and every such detailed paper is helpful.

The manuscript is well written and very interesting. `I am particularly interested in the paper as I had a patient with heart tumor last week and our team had several concern in the management. Such case report is therefore much appreciated, especially since there is scarce information on such patients in the literature.
the only issue would be to describe wider the complications after surgery they observed. No further controls are possible as the patient who was described, died.
I do suggest approvement of the case. It is of diagnostic vale.

Author Response

Response to Reviewer 2 Comments

Point 1: The paper presents a rare however very important issue of angiosarcoma of the heart. There is scarce information on the diagnostics and treatment of such rare cases in the literature, and every such detailed paper is helpful.
The manuscript is well written and very interesting. `I am particularly interested in the paper as I had a patient with heart tumor last week and our team had several concern in the management. Such case report is therefore much appreciated, especially since there is scarce information on such patients in the literature.
the only issue would be to describe wider the complications after surgery they observed. No further controls are possible as the patient who was described, died.
I do suggest approvement of the case. It is of diagnostic vale.

Response 1: I have added a descripition of more detailed clinical course of the patient’s death (Line 69-75).